# Teaching LLMs According to Their Aptitude: Adaptive Switching Between CoT and TIR for Mathematical Problem Solving

## Abstract

Existing supervised fine-tuning (SFT) approaches to enhance the mathematical reasoning of large language models (LLMs) rely either on Chain-of-Thought (CoT) for generalizability or Tool-Integrated Reasoning (TIR) for precise computation. While efforts have been made to combine these methods, they primarily rely on post-selection or predefined strategies, leaving an open question: Could we endow LLMs with the ability to adaptively determine whether to use CoT or TIR based on the math problems at hand before decoding? In this work, we propose **TATA** (**T**eaching LLMs **A**ccording to **T**heir **A**ptitude), an adaptive framework that enables LLMs to personalize their reasoning strategy for different problems spontaneously, aligning it with their intrinsic aptitude. TATA incorporates base-LLM-aware data selection during SFT to tailor training data to the model's unique abilities, which equips LLMs to autonomously determine and apply the effective reasoning strategy at test time. Empirical results demonstrate that TATA effectively combines the complementary strengths of CoT and TIR, achieving superior or comparable performance with improved inference efficiency compared to existing methods. Further analysis highlights the crucial role of aptitude-aware data selection in enabling LLMs to make informed and adaptive reasoning decisions, aligning reasoning strategies with model capabilities.

## 1 Introduction

Previous SFT methods for mathematical reasoning (Tong et al., 2024; Shao et al., 2024; Yan et al., 2024; Gou et al., 2023; Wang et al., 2023; Lu et al., 2024) predominantly adopt one of the following two distinct reasoning paradigms: Chain-of-Thought (CoT) reasoning (Wei et al., 2022) or Tool-Integrated Reasoning (TIR) (Chen et al., 2022; Gao et al., 2023). CoT employs natural language (NL) to articulate intermediate reasoning steps, whereas TIR integrates NL with Python code blocks in an interleaved manner (see Section 3.2). While CoT offers computational efficiency, it may compromise the numerical accuracy of complex calculations. In contrast, TIR's structured execution of code ensures precise computation but incurs significant computational overhead. Notably, recent studies (Zhao et al., 2023; Yang et al., 2024b) have empirically demonstrated that CoT and TIR exhibit complementary strengths: CoT demonstrates superior performance on problems demanding sophisticated logical deduction with minimal numerical computation, whereas TIR excels in scenarios requiring intensive numerical calculations with relatively simpler logical flow.

This complementary nature suggests potential benefits to integrate these two reasoning patterns. Zhao et al. (2023) proposes an auxiliary LLM-based selector to dynamically choose between paradigms via prompt-based routing (Figure 1 (a)). MAmmoTH (Yue et al., 2023) switches to CoT reasoning if TIR encounters execution errors or timeouts (Figure 1 (b)). Yang et al. (2024b) employs different inference prompts to elicit respective reasoning capabilities (Figure 1 (c)). Despite these advancements, existing approaches predominantly rely on either external selectors (as in Zhao et al. (2023)) or predefined heuristics (as in MAmmoTH and Qwen-2.5-Math) rather than endowing LLMs with the intrinsic capability to autonomously recognize appropriate reasoning strategies. However, the potential for LLMs themselves to dynamically adapt reasoning paradigms (CoT or TIR) remains underexplored.

To bridge this gap, we propose **T**eaching LLMs **A**ccording to **T**heir **A**ptitude (**TATA**), an adaptive framework that enables LLMs to spontaneously select between CoT and TIR for math problem solving. Instead of adopting a fixed strategy for all training queries, TATA adaptively tailors the training data selection process by considering both the query characteristics and the base LLM's aptitude. This ensures that the resulting model is equipped to select a suitable reasoning strategy (CoT or TIR) for different queries at test time, facilitating aptitude-driven reasoning. As a result, TATA preserves and enhances the generalizability of the model, particularly for out-of-domain tasks.

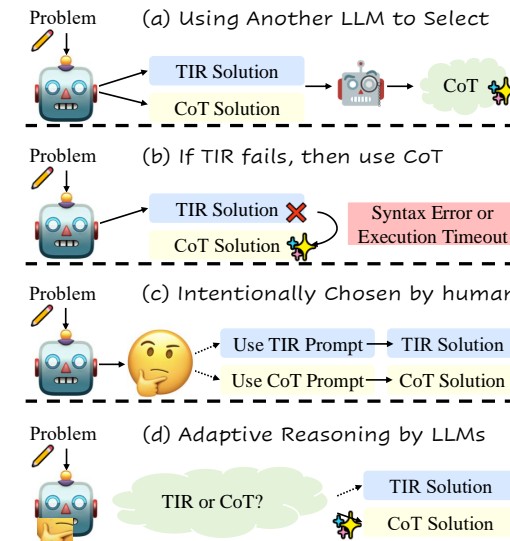

Figure 1: Illustration of our research question. (a) Zhao et al. (2023) post-select between CoT and TIR by another LLM. (b) Yue et al. (2023) choose CoT if TIR fails due to syntax error or execution timeout. (c) Yang et al. (2024a) controls selection between CoT and TIR by predefined inference prompts. (d) We aim to teach LLMs to choose the appropriate one before decoding.

Concretely, we begin with a dataset $\mathcal{D}$, which consists of $N$ triplets, each containing a query, a CoT solution, and a TIR solution. We then construct an anchor set, $\mathcal{D}_{\text{anchor}}$, to evaluate the model's performance. For each training query in $\mathcal{D}$, we assess the LLM's accuracy on $\mathcal{D}_{\text{anchor}}$ by providing either the CoT or TIR solution of the query as a one-shot example. Based on the model's performance on the $\mathcal{D}_{\text{anchor}}$ in each setting, we select the most effective reasoning paradigm for training queries and use it to construct the SFT data, $\mathcal{D}_{\text{SFT}}$. We endow the base LLMs with the ability to adaptively switch between CoT and TIR by training of personalized training set $\mathcal{D}_{\text{SFT}}$. To assess TATA's effectiveness, we conduct extensive evaluations across six math reasoning benchmarks, utilizing both general-purpose LLMs (e.g. Llama-3-8B (AI@Meta, 2024)) and math-specialized LLMs (e.g. Qwen2.5-Math-7B) as base models. Experiments show that TATA successfully leads to better performance across various models and benchmarks.

To summarize, our contributions are as follows:

1. We propose TATA, an adaptive framework that enables LLMs to spontaneously select between CoT and TIR for adaptive mathematical reasoning based on their inherent aptitudes.

2. Extensive experiments demonstrate that TATA effectively combines the strengths of both CoT and TIR, achieving comparable or even superior performance while offering higher inference efficiency compared to TIR.

3. Comprehensive analyses highlight the critical role of base-LLM-aware data selection for CoT and TIR, which is the core of our TATA framework.

## 2 RELATED WORK

**Math Reasoning with CoT and TIR** CoT and TIR are two widely recognized approaches for reasoning with LLMs. CoT offers interpretability and generalizability, while TIR can provide precise calculation results. Previous work on mathematical SFT has primarily focused on either CoT (Yu et al., 2023; Tong et al., 2024; Shao et al., 2024; Yan et al., 2024) or TIR (Yue et al., 2023; Gou et al., 2023; Wang et al., 2023; Yin et al., 2024), with a few efforts to integrate both (Yue et al., 2023; Beeching et al., 2024; Yang et al., 2024b). For instance, MAmmoTH (Yue et al., 2023) mainly adopts TIR but switches to CoT when code execution fails due to errors or timeouts. However, it relies on separate prompts and manual inference controls to switch between them. Recent work has explored automatic selection between CoT and TIR (Zhao et al., 2023; Yue et al., 2024; Yu et al., 2024), such as using an auxiliary LLM to determine CoT/TIR (Zhao et al., 2023). However, these methods rely

on external planners to select CoT/TIR, not by LLMs themselves. In contrast, our work seeks to enable LLMs to spontaneously select the appropriate reasoning strategy without relying on external planners or manual interventions.

**Data Selection** Data selection plays a crucial role in training LLMs (Albalak et al., 2024). Various methods have been developed to optimize data usage at different stages of model training, ranging from pretraining (Brown et al., 2020; Wettig et al., 2024; Lin et al., 2025) to supervised fine-tuning (SFT) (Li et al., 2023; Pan et al., 2024; Xia et al., 2024; Zhou et al., 2023b). Our work focuses specifically on data selection between CoT and TIR given a math problem and a base LLM.

**Test-Time Scaling** Recent efforts in scaling test-time computation have explored refinement strategies (Snell et al., 2024; Xu et al., 2024b; Hou et al., 2025; Lee et al., 2025), which iteratively build on previous outputs, and MCTS-based approaches (Zhou et al., 2023a; Liu et al., 2024; Wu et al., 2024). The roles of SFT and RL have also been actively discussed (Chu et al., 2025). For example, OpenAI (2024); DeepSeek-AI et al. (2025) use RL to train LLMs for generating longer CoT reasoning, while Muennighoff et al. (2025); Ye et al. (2025) leverage SFT for scaling test-time computation. This work focuses on enabling adaptive mathematical reasoning in LLMs primarily through data selection during the SFT stage, with discussions on the potential use of RL in Section 6.3. While existing test-time scaling methods mainly target CoT, exploring adaptive selection between CoT and TIR could be an orthogonal direction.

## 3 BACKGROUND

### 3.1 REJECTION FINE-TUNING

Rejection fine-tuning (RFT) is a widely-adopted approach to enhance math reasoning abilities by augmenting the original training set using rejection sampling (Yuan et al., 2023). Suppose that the original training set $\mathcal{D}_{\text{orig}} = \{(x_i, y_i)\}_{i=1}^{N}$ consists of N pairs of data points $(x_i, y_i)$. For each query $x_i$, M responses are generated by a teacher model (e.g., GPT-4): $\{x_i, y_i^j\}_{j=1}^{M}$. If $y_i^j \neq y_i$, then the response $y_i^j$ is discarded, leading to the augmented training set $\mathcal{D}_{\text{aug}} = \{(x_i, y_i^j)\}_{i=1}^{N} {}_{j=1}^{M_i}$, where $M_i \leq M$ is the number of correct responses for query $x_i$. More details are given in Appendix A.1.

### 3.2 TIR INFERENCE PIPELINE

Tool-Integrated Reasoning (TIR) (Gou et al., 2023) combines natural language reasoning with Python code execution in an interleaved manner. When a Python code block is encountered, it is executed using a Python interpreter, and the resulting output, along with the previous context, is fed back into the LLM to facilitate further reasoning (see Algorithm 1). Solving math problems with TIR often requires multiple iterations of these interactions, which typically results in higher computational costs compared to CoT. However, TIR offers more reliable results by leveraging external tools for computation. The whole inference pipeline of TIR is provided in Appendix A.2.

### 3.3 IMPLICIT INSTRUCTION TUNING

In-Context Learning (ICL) can be viewed as implicit instruction tuning (IIT), i.e., "fine-tune" the demonstration implicitly (Li et al., 2023). Let $\mathbf{X}_{\text{ins}}, \mathbf{X}_{\text{test}} \in \mathbb{R}^{d_{\text{in}}}$ be the few-shot demonstration inputs and the test input, respectively. Suppose $\mathbf{W}_K, \mathbf{W}_V, \mathbf{W}_Q \in \mathbb{R}^{d_{\text{out}} \times d_{\text{in}}}$ are projection matrices to compute the attention queries, keys, and values. The self-attention is formulated as follows:

$$\mathbf{W}_V[\mathbf{X}_{\text{ins}} \| \mathbf{X}_{\text{test}}] \mathsf{Softmax}\left(\frac{\mathbf{W}_K[\mathbf{X}_{\text{ins}} \| \mathbf{X}_{\text{test}}]^\top \boldsymbol{Q}}{\sqrt{d_{\text{in}}}}\right)$$
$$\approx [\underbrace{\mathbf{W}_V \mathbf{X}_{\text{test}}(\mathbf{W}_K \mathbf{X}_{\text{test}})^\top}_{\textit{Only test input.}} + \underbrace{\mathbf{W}_V \mathbf{X}_{\text{ins}}(\mathbf{W}_K \mathbf{X}_{\text{ins}})^\top}_{\textit{Only instruction sample.}}]\boldsymbol{Q},$$

where $\|$ denotes concatenation. The first term only involves the test input $\mathbf{X}_{\text{test}}$ and the second term is related to few-shot exemplars, which can be interpreted as an IIT to the model parameters (Dai et al., 2022; Yang et al., 2023) (see Appendix A.3).

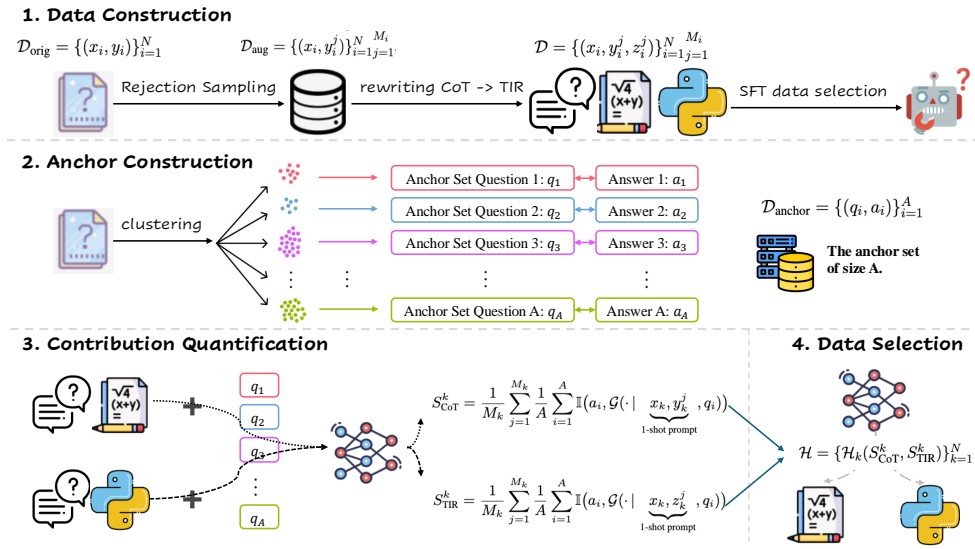

Figure 2: Overview of our **T**eaching LLMs **A**ccording to **T**heir **A**ptitude (**TATA**) framework. Here, $\mathcal{D}_{\text{orig}}$ denotes the original training set, $\mathcal{D}_{\text{aug}}$ represents the augmented training set obtained through rejection sampling with CoT only, and $\mathcal{D}$ refers to the candidate set consisting of (query, CoT, TIR) triplets. $\mathcal{D}_{\text{anchor}}$ is the anchor set of size $A$. $S_{\text{CoT}}^k$ and $S_{\text{TIR}}^k$ are scores calculated based on the LLMs' aptitude on the anchor set, elicited using 1-shot prompts. Finally, $\mathcal{H}$ represents the SFT data selection process. **Fine-tuning on the resulting SFT data enables LLMs to spontaneously select between CoT and TIR at test time according to their aptitude.**

## 4 THE TATA FRAMEWORK

### 4.1 PROBLEM SETTING

In this section, we formally formulate our problem setting, including our data structure and objective.

**Data Structure**   Suppose we have a candidate dataset $\mathcal{D} = \{(x_i, y_i^j, z_i^j)\}_{i=1}^{N}{}_{j=1}^{M_i}$ consisting of triplets in the form $(x_i, y_i^j, z_i^j)$ for the $i$-th training example, where $1 \le j \le M_i$. Here, $x_i$ represents the $i$-th training problem, while $y_i^j$ and $z_i^j$ denote the $j$-th CoT solution and TIR solution to this problem, respectively. Notably, the TIR solution $z_i^j$ is adapted from $y_i^j$, meaning both solutions follow the same steps to solve the mathematical problem $x_i$, but differ in their reasoning formats: $y_i^j$ relies exclusively on natural language reasoning, whereas $z_i^j$ incorporates Python code blocks to perform calculations for certain reasoning steps.

**Objective**   Our objective is to construct an SFT dataset from the candidate dataset $\mathcal{D} = \{(x_i, y_i^j, z_i^j)\}_{i=1}^{N}{}_{j=1}^{M_i}$ by incorporating suitable reasoning patterns for different training queries. Specifically, for each problem $x_i$ in $\mathcal{D} = \{(x_i, y_i^j, z_i^j)\}_{i=1}^{N}{}_{j=1}^{M_i}$, we need to decide whether to include its CoT solutions or TIR solutions in the SFT dataset. Formally, this involves determining whether $\{(x_i, y_i^j)\}_{j=1}^{M_i} \subseteq D_{\text{SFT}}$ or $\{(x_i, z_i^j)\}_{j=1}^{M_i} \subseteq D_{\text{SFT}}$.[1] For example, CoT-only SFT (Xu et al., 2024c) constructs the dataset such that $\{(x_i, y_i^j)\}_{j=1}^{M_i} \subseteq D_{\text{SFT}}, \forall i$. In contrast, TIR-only SFT (Gou et al., 2023) selects $\{(x_i, z_i^j)\}_{j=1}^{M_i} \subseteq D_{\text{SFT}}, \forall i$. Unlike these static selection approaches, TATA aims to dynamically tailor the most suitable reasoning paradigm for different training queries and base LLMs.

---

[1]We also consider scenarios where both CoT and TIR solutions for a query are included in the SFT dataset.

## 4.2 TATA OVERVIEW

*"Teach according to students' aptitude."*

— Confucius

**Motivation**  Intuitively, if an LLM demonstrates improved performance on certain queries when fine-tuned with CoT solutions instead of TIR solutions, it suggests its inclination toward CoT reasoning in those cases. This preference can be extrapolated to new cases, where the model is expected to favor CoT for similar problems during testing. The same principle applies to TIR-based reasoning. Inspired by IIT theory (see Section 3.3), LLMs can be indirectly "fine-tuned" with CoT or TIR examples through one-shot learning, thereby replacing the need for actual SFT.

**Overview**  As depicted in Figure 2, our proposed framework, TATA, comprises four main steps: data construction, anchor construction, contribution quantification, and data selection. In the data construction stage, we adapt an original training set, $\mathcal{D}_{\text{orig}}$, containing CoT solutions, to form the candidate set $\mathcal{D} = \{(x_i, y_i^j, z_i^j)\}_{i=1}^{N}{}_{j=1}^{M_i}$. This candidate set includes triplets of queries, a CoT solution, and corresponding TIR solution. Next, during the anchor construction stage, a representative anchor set of size $A$ is generated from the original training set by clustering. In the contribution quantification stage, we compute two scores, $S_{\text{CoT}}^k$ and $S_{\text{TIR}}^k$, for each query $q_k$ in the candidate set $\mathcal{D} = \{(x_i, y_i^j, z_i^j)\}_{i=1}^{N}{}_{j=1}^{M_i}$. These scores indicate the impact of CoT and TIR solutions on the performance of LLMs using IIT (see Section 3.3). The data selection step formulates a decision based on $S_{\text{CoT}}^k$ and $S_{\text{TIR}}^k$, determining whether to include CoT or TIR solutions for queries in $\mathcal{D}$. Finally, SFT is performed on this curated training set.

## 4.3 TATA DETAILS

**Data Construction**  We start with an original math training set (e.g., MATH (Hendrycks et al., 2021) training set), denoted as $\mathcal{D}_{\text{orig}} = \{(x_i, y_i)\}_{i=1}^{N}$, which consists of $N$ training examples, where the $i$-th problem is represented as $x_i$ with its corresponding golden answer $y_i$. To further enhance the training set, we apply RFT (see Section 3.1), resulting in an augmented dataset, $\mathcal{D}_{\text{aug}} = \{(x_i, y_i^j)\}_{i=1}^{N}{}_{j=1}^{M_i}$, where $y_i^j$ denotes the $j$-th augmented CoT solution for the $i$-th training problem $x_i$. Next, we convert each CoT solution $y_i^j$ into the TIR format $z_i^j$ by prompting a strong LLM (e.g., GPT-4o). During this process, the original logic in $y_i^j$ is preserved, while Python blocks are introduced to handle necessary computations. This transformation produces a candidate dataset $\mathcal{D} = \{(x_i, y_i^j, z_i^j)\}_{i=1}^{N}{}_{j=1}^{M_i}$, which is required for our problem setting (see Section 4.1).

**Anchor Construction**  To evaluate the impact of specific CoT or TIR solutions on the performance of LLMs, we construct an anchor set, denoted by $\mathcal{D}_{\text{anchor}} = \{(q_i, a_i)\}_{i=1}^{A}$, where $A$ is the size of the anchor set, $q_i, a_i$ is the $i$-th question and corresponding ground-truth answer in $\mathcal{D}_{\text{anchor}}$. We expect $\mathcal{D}_{\text{anchor}}$ to be diverse, ensuring that accuracy on this set fairly reflects the LLMs' overall performance. To achieve this, we first encode all queries from $\mathcal{D}_{\text{orig}}$ into vector representations using an embedding model (e.g., text-embedding-ada-002) and then cluster them into $A$ distinct groups. The center of each cluster is selected to $\mathcal{D}_{\text{anchor}}$. This approach takes the semantic diversity of questions into account, making $\mathcal{D}_{\text{anchor}}$ a reliable indicator of LLMs' performance. To put it simply, one can treat this $\mathcal{D}_{\text{anchor}}$ as a validation set to validate the performance of a base model in different settings.

**Contribution Quantification**  To quantify the contribution of CoT and TIR for each triplet $(x_k, y_k^j, z_k^j)$ in $\mathcal{D}$ to the LLMs' math reasoning abilities, we implicitly "fine-tune" the LLMs using CoT and TIR formats separately through one-shot learning (see Section 3.3). In this case, the performance of the base model under one-shot ICL approximates the accuracy achieved by a model that is finetuned from the same base model using the same one-shot example. For the $k$-th query $x_k$ and its corresponding CoT solutions $y_k^j$ ($1 \leq j \leq M_k$), we compute a CoT score, denoted as $S_{\text{CoT}}^k$, as follows:

$$S_{\text{CoT}}^k = \frac{1}{M_k} \sum_{j=1}^{M_k} \frac{1}{A} \sum_{i=1}^{A} \mathbb{I}\big(a_i, \mathcal{G}(\cdot \mid \underbrace{x_k, y_k^j}_{\text{1-shot prompt}}, q_i)\big),$$

Table 1: The accuracies (%) of our TATA framework, comparing with various baselines. The best accuracies within each group are shown in **bold**. "ID AVG", "OOD AVG", and "AVG" denote the averages of these metrics across in-domain, out-of-domain, and all six benchmarks.

| Model | Method | In-Domain | | | Out-of-Domain | | | | | AVG |
|---|---|---|---|---|---|---|---|---|---|---|
| | | GSM8K | MATH | ID AVG | MAWPS | SVAMP | College | Olympiad | OOD AVG | |
| Qwen2.5-0.5B | hybrid | 49.3 | 37.7 | 43.5 | 84.5 | 55.0 | **27.5** | 7.9 | 43.7 | 43.6 |
| | ensemble | 47.1 | 34.8 | 41.0 | 83.4 | 53.8 | 25.6 | 7.7 | 42.6 | 42.1 |
| | GPT-Select | 45.6 | 31.6 | 38.6 | 80.4 | 52.6 | 24.4 | 7.1 | 41.1 | 40.3 |
| | TATA | **52.8** | 36.6 | **44.7** | **85.9** | **59.4** | 26.9 | **8.6** | **45.2** | **45.0** |
| Qwen2.5-1.5B | hybrid | 71.3 | **54.7** | 63.0 | 91.8 | 80.4 | 36.8 | **19.7** | 57.2 | 59.1 |
| | ensemble | 71.1 | 54.3 | 62.7 | 91.5 | 79.6 | 36.6 | 18.8 | 56.6 | 58.7 |
| | GPT-Select | 72.5 | 47.3 | 59.9 | 91.8 | 81.8 | 35.0 | 14.8 | 55.8 | 57.2 |
| | TATA | **77.6** | 53.8 | **65.7** | **94.2** | 80.7 | **37.0** | 18.8 | **57.7** | **60.4** |
| Qwen2.5-3B | hybrid | 80.9 | **61.9** | 71.4 | 90.2 | 79.8 | 41.6 | 24.4 | 59.0 | 63.1 |
| | ensemble | 81.3 | 60.3 | 70.8 | **95.3** | **86.2** | **42.9** | 23.1 | 61.9 | 64.8 |
| | GPT-Select | 81.4 | 53.6 | 67.5 | 86.2 | 79.0 | 38.9 | 17.3 | 33.8 | 45.0 |
| | TATA | **84.0** | 61.3 | **72.6** | 94.7 | 85.3 | 41.6 | **24.9** | **61.6** | **65.3** |
| Qwen2.5-7B | hybrid | 87.0 | **67.5** | 77.3 | 92.1 | 84.3 | **44.2** | **31.7** | 63.1 | 67.8 |
| | ensemble | 87.1 | 63.0 | 75.0 | 91.5 | 82.0 | 43.0 | 30.2 | 61.7 | 66.1 |
| | GPT-Select | 88.3 | 59.0 | 73.7 | 91.4 | 83.4 | 42.7 | 23.3 | 60.2 | 64.7 |
| | TATA | **89.5** | 66.8 | **78.2** | 94.2 | **86.2** | 43.4 | 31.1 | **63.7** | **68.5** |
| Qwen2.5-14B | hybrid | 91.4 | **71.7** | 81.5 | 93.8 | 84.5 | 45.8 | **35.3** | 64.8 | 70.4 |
| | ensemble | 90.1 | 66.9 | 78.5 | 92.2 | 82.8 | 46.1 | 32.3 | 63.3 | 68.4 |
| | GPT-Select | 90.7 | 61.5 | 76.1 | 86.2 | 79.1 | 44.1 | 23.0 | 58.1 | 64.1 |
| | TATA | **92.1** | **71.7** | **81.9** | **96.5** | **88.4** | **46.4** | **35.3** | **66.7** | **71.7** |
| LLaMA-3-8B | hybrid | 82.0 | **56.1** | 69.1 | 88.0 | 78.0 | 30.8 | 21.3 | 54.5 | 59.4 |
| | ensemble | **84.0** | 46.9 | 65.4 | 88.6 | 79.3 | 29.6 | 15.3 | 53.2 | 57.3 |
| | GPT-Select | 83.2 | 47.2 | 65.2 | 85.3 | 77.5 | 30.6 | 13.9 | 51.8 | 56.3 |
| | TATA | **84.0** | 55.1 | **69.6** | **91.8** | **82.7** | **34.2** | **21.5** | **57.6** | **61.5** |
| Qwen2.5Math-1.5B | hybrid | 82.6 | **66.3** | **74.4** | 92.7 | 83.6 | 43.1 | 26.2 | 61.4 | 65.7 |
| | ensemble | 81.5 | 64.7 | 73.1 | 91.8 | 83.9 | 44.1 | **27.4** | 61.8 | 65.6 |
| | GPT-Select | 79.4 | 56.9 | 68.1 | 92.7 | 83.7 | 41.8 | 20.6 | 59.7 | 62.5 |
| | TATA | **83.2** | 62.8 | 73.0 | **94.0** | **85.6** | 43.9 | 26.8 | **62.6** | **66.0** |
| Qwen2.5Math-7B | hybrid | 89.2 | **73.4** | 81.3 | 95.4 | **89.5** | 47.1 | 34.4 | 66.6 | 71.5 |
| | ensemble | 89.1 | 67.7 | 78.4 | 93.4 | 84.5 | 46.7 | 30.8 | 63.9 | 68.8 |
| | GPT-Select | 89.8 | 63.0 | 76.4 | 89.4 | 85.1 | 44.4 | 24.6 | 60.7 | 65.9 |
| | TATA | **89.8** | 73.0 | **81.4** | 95.2 | 88.1 | **48.3** | **35.9** | **66.9** | **71.7** |

where $x_k$ and $y_k^j$ serve as the one-shot prompt for the LLM $\mathcal{G}$ to generate a response for the question $q_i$ in the anchor set, and $\mathbb{I}$ is an indicator function that returns 1 if the model's generated answer matches the ground-truth answer $a_i$ of question $q_i$, and 0 otherwise. $S_{\text{CoT}}^k$ represents the average accuracy on the anchor set $\mathcal{D}_{\text{anchor}}$ when using CoT format as the one-shot prompt, averaged over all CoT solutions $y_k^j$ ($1 \leq j \leq M_k$) for query $x_k$. Similarly, the TIR score, $S_{\text{TIR}}^k$, is defined as:

$$ S_{\text{TIR}}^k = \frac{1}{M_k} \sum_{j=1}^{M_k} \frac{1}{A} \sum_{i=1}^{A} \mathbb{I}\big(a_i, \mathcal{G}(\cdot \mid \underbrace{x_k, z_k^j}_{\text{1-shot prompt}}, q_i)\big). $$

The only difference is that the TIR format $z_k^j$ is used as the one-shot example instead of CoT.

**Data Selection**   Currently, two scores, $S_{\text{CoT}}^k$ and $S_{\text{TIR}}^k$, are associated with the $k$-th query $q_k$ in the candidate set $\mathcal{D}$. The next step is to determine whether to include the CoT or the TIR solutions for this specific query $q_k$ in $\mathcal{D}$. Specifically, the goal is to decide between $\{(x_k, y_k^j)\}_{j=1}^{M_k} \subseteq D_{\text{SFT}}$ or $\{(x_k, z_k^j)\}_{j=1}^{M_k} \subseteq D_{\text{SFT}}$. We formalize this decision process with a decision function $\mathcal{H}_k = (S_{\text{CoT}}^k, S_{\text{TIR}}^k)$, where the final decision is represented as a series of decisions $\mathcal{H} = \{\mathcal{H}_k\}_{k=1}^{N}$, where $N$ is the number of queries in candidate set $\mathcal{D}$. For instance, a simple decision function $\mathcal{H}_k$ could involve consistently choosing CoT solutions, i.e., $\{(x_k, y_k^j)\}_{j=1}^{M_k} \subseteq D_{\text{SFT}}$ for all $k$. This corresponds to performing SFT exclusively on CoT data.

Table 2: Ablation of Contribution Quantification.

| Model | Method | In-Domain | | | Out-of-Domain | | | | | AVG |
|---|---|---|---|---|---|---|---|---|---|---|
| | | GSM8K | MATH | ID AVG | MAWPS | SVAMP | College | Olympiad | OOD AVG | |
| | hybrid | 49.3 | 37.7 | 43.5 | 84.5 | 55.0 | 27.5 | 7.9 | 43.7 | 43.6 |
| | ensemble | 47.1 | 34.8 | 41.0 | 83.4 | 53.8 | 25.6 | 7.7 | 42.6 | 42.1 |
| | GPT-Select | 45.6 | 31.6 | 38.6 | 80.4 | 52.6 | 24.4 | 7.1 | 41.1 | 40.3 |
| Qwen2.5-0.5B | CoT+TIR | 51.5 | 33.5 | 42.5 | 85.8 | 58.6 | 25.7 | 7.9 | 44.4 | 43.8 |
| | TATA - random 100 | 50.6 | 34.6 | 42.6 | 85.7 | 57.6 | 26.2 | 6.2 | 43.9 | 43.5 |
| | TATA - A 200 | 52.6 | 36.8 | 44.7 | 85.1 | 59.6 | 27.4 | 8.4 | 45.1 | 45.0 |
| | TATA | **52.8** | **36.6** | **44.7** | **85.9** | **59.4** | **26.9** | **8.6** | **45.2** | **45.0** |

# 5 EXPERIMENTAL RESULTS

## 5.1 EXPERIMENTAL SETUP

**TATA Implementation**   We select the training sets from GSM8K (Cobbe et al., 2021) and Math (Hendrycks et al., 2021) as $\mathcal{D}_{orig}$. For $\mathcal{D}_{aug}$, we use the DART-Math-Hard dataset (Tong et al., 2024). We employ GPT-4o to rewrite CoT solutions into TIR format using carefully curated prompts and filter out triplets with anomalous TIR responses (e.g., those that lack a definitive conclusion regarding the final answer). For embedding, we use text-embedding-ada-002 to encode all queries in $\mathcal{D}$ into 1,536-dimensional vectors. We set the size of $\mathcal{D}_{anchor}$ to 100 for both the GSM8K and Math. To save computational cost, we randomly sample one pair of CoT and TIR solutions per candidate query, leading to a new candidate set, $\mathcal{D}^* = \{(x_i, y_i^*, z_i^*)\}_{i=1}^{N}$. For the decision function $\mathcal{H}$, we determine selection criteria based on two quantiles of the distribution of $(S_{CoT} - S_{TIR})$. More details are provided in Appendix B.1.

**Evaluation Benchmarks**   We evaluate our approach using six benchmarks for both in-domain and out-of-domain (OOD) assessment. Specifically, we use the GSM8K and MATH test sets for in-domain evaluation. For OOD evaluation, we include the SVAMP (Patel et al., 2021), MAWPS (Koncel-Kedziorski et al., 2016), CollegeMath (Tang et al., 2024), and OlympiadBench-Math (He et al., 2024) (details in Appendix B.2)

**Evaluation Metrics**   In addition to measuring accuracy on various benchmarks, we evaluate the generation time cost using the average number of total tokens per generation and quantify the cost of invoking Python interpreters by the average number of code executions (see Appendix B.3).

**Baselines**   We include the following methods as our baselines: 1) *Hybrid* (Yue et al., 2023): Primarily uses TIR but falls back to CoT upon code execution errors or timeouts (Figure 1 (b)). 2) *Ensemble* (Zhao et al., 2023): Post-selects between TIR and CoT outputs using an additional LLM (Figure 1 (a)). In our implementation, we use the same 8-shot prompt as Zhao et al. (2023) with the base LLM as the selector for consistency. 3) *GPT-Select*: Uses GPT-4o during data selection to choose CoT or TIR per query, testing whether a strong external LLM can effectively select reasoning paradigms regardless of the base LLM's aptitude.

Additional details, including the SFT setup and evaluation setup, are provided in Appendix B.4.

## 5.2 MAIN RESULTS

**Effectiveness of TATA**   Results presented in Table 1 demonstrate the effectiveness of our proposed TATA framework. Across various base models, model sizes, and benchmarks, TATA consistently achieves competitive or superior performance compared to all the other baselines, highlighting its ability to leverage the complementary advantages of both methods. Additionally, TATA achieves significantly better performance than the "GPT-Select" baseline. While "GPT-Select" leverages a much stronger LLM to select between CoT and TIR for different queries, it demonstrates that this approach may not be suitable for all base LLMs. This highlights the critical importance of base-LLM-aware selection in optimizing performance.

**Inference efficiency**   The results in Table 3 demonstrate that our TATA not only improves accuracy but also enhances inference efficiency compared to standalone CoT and TIR methods. Across all

model sizes, TATA achieves higher accuracy while maintaining lower token usage and fewer code executions than TIR, and it significantly reduces computational overhead compared to TIR without sacrificing the benefits of tool integration. For instance, with Qwen2.5-7B, TATA achieves a 2.3% accuracy improvement over CoT while using 9.1 fewer tokens per generation and only 1.4 code executions, compared to TIR's 2.63 code executions. This balance between accuracy and efficiency highlights TATA's ability to streamline reasoning processes, making it a computationally effective solution for mathematical reasoning tasks. The "hybrid" and "ensemble" approaches incur even higher inference costs compared to our proposed TATA. Specifically, "hybrid" requires decoding via TIR and selectively switching to CoT execution for specific cases; "ensemble" generates both CoT and TIR outputs during testing and incurs additional costs for selection between the two.

## 5.3 ABLATION

Table 4: TATA is not sensitive to quantiles. * denotes the quantiles we choose for Qwen2.5Math-0.5B.

| Quantiles | 50, 60 | 40, 60 | 30, 60 | 30, $65^*$ | 30, 70 |
|---|---|---|---|---|---|
| AVG | 44.8 | 44.8 | 44.9 | **45.0** | 44.8 |

**Quantile selection** As mentioned in Section 5.1, the data selection function $\mathcal{H}$ is determined using two quantiles of the distribution $(S_{\text{CoT}}^k - S_{\text{TIR}}^k)$ (see Appendix B). These quantiles are selected through the grid search. As shown in Table 4, the performance of TATA is not very sensitive to the choice of these quantiles (see Appendix B).

**Anchor set & Others** Table 2 includes results for several other ablation studies: 1) "CoT + TIR": This method includes all CoT and TIR solutions for each query without any data selection. 2) Anchor set construction with random sampling ("TATA - random 100"): Replacing k-means clustering with random selection while keeping the anchor set size constant. 3) Larger anchor set size ("TATA - A=200"): Increasing the anchor set size to 200. From Table 2, we observe that TATA achieves the highest overall accuracy. Naively including all CoT and TIR solutions (i.e., "CoT + TIR") results in a notice-

Table 3: Results of inference costs. The three metrics, "Acc", "Token", and "# Code" represent the average accuracy (%), total tokens per generation, and number of code executions.

| Model | Method | Acc↑ | Token↓ | # Code↓ |
|---|---|---|---|---|
| Qwen2.5-3B | TATA | 65.3 | 383.4 | 1.43 |
| | CoT | $62.9_{-2.4}$ | $385.2_{+1.8}$ | $0_{-1.43}$ |
| | TIR | $62.9_{-2.4}$ | $411.3_{+27.9}$ | $2.8_{+1.37}$ |
| Qwen2.5-7B | TATA | 68.5 | 369.1 | 1.4 |
| | CoT | $66.2_{-2.3}$ | $378.2_{+9.1}$ | $0_{-1.40}$ |
| | TIR | $67.8_{-0.7}$ | $393.2_{+24.1}$ | $2.63_{+1.23}$ |
| LLaMA-3-8B | TATA | 61.5 | 371.7 | 1.32 |
| | CoT | $58_{-3.5}$ | $386_{+14.3}$ | $0_{-1.32}$ |
| | TIR | $59.3_{-2.2}$ | $392.5_{+20.8}$ | $2.66_{+1.34}$ |
| Qwen2.5Math-1.5B | TATA | 66.0 | 405.4 | 1.08 |
| | CoT | $63.4_{-2.6}$ | $388.5_{+16.9}$ | $0_{-1.08}$ |
| | TIR | $64.8_{-1.2}$ | $460.1_{+54.7}$ | $3.23_{+2.15}$ |
| Qwen2.5Math-7B | TATA | 71.7 | 393.8 | 1.26 |
| | CoT | $67.5_{-4.2}$ | $379.9_{+13.9}$ | $0_{-1.26}$ |
| | TIR | $71.6_{-0.1}$ | $417.8_{+24.0}$ | $2.68_{+1.42}$ |

able decline in performance, despite the larger size of the $\mathcal{D}_{\text{SFT}}$ dataset. Random anchor set selection ("TATA - random 100") critically degrades performance, highlighting the importance of a representative anchor set over size alone. Increasing the anchor set size shows diminishing returns, indicating that $A = 100$ is enough for model evaluation in our SFT data curation.

## 6 ANALYSIS AND DISCUSSION

### 6.1 ANALYSIS OF COT SCORES AND TIR SCORES

To further investigate how different LLMs exhibit varying reasoning patterns, we analyze the distribution of $S_{\text{CoT}}^k$ and $S_{\text{TIR}}^k$. As illustrated in Figure 3 (see also Appendix C.2), different base LLMs display distinct distributions of $(S_{\text{CoT}}^k - S_{\text{TIR}}^k)$, indicating varying inclinations towards CoT and TIR reasoning for queries in the candidate set $\mathcal{D}^* = \{(x_i, y_i^*, z_i^*)\}_{i=1}^N$. Interestingly, even base LLMs from the same family can demonstrate different tendencies towards CoT and TIR (e.g., Qwen2.5-0.5B vs. Qwen2.5-7B). Notably, Qwen2.5-7B exhibits a stronger preference for CoT on GSM8K and for TIR on MATH, compared to Qwen2.5-0.5B.

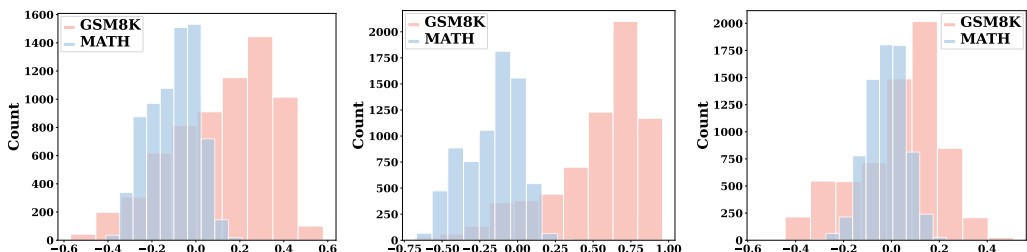

Figure 3: The distribution of $(S_{\text{CoT}}^k - S_{\text{TIR}}^k)$ for GSM8K (red) and MATH (blue): Qwen2.5-0.5B (left), Qwen2.5-7B (middle), LLaMA-3-8B (right).

## 6.2 TRANSFERABILITY OF DATA SELECTION BETWEEN DIFFERENT LLMS

To evaluate whether data selected by one LLM can benefit another LLM, we conducted additional experiments using Qwen2.5-0.5B to assess this type of transferability. Specifically, we fine-tuned Qwen2.5-0.5B on data selected by Qwen2.5-7B and LLaMA-3-8B, with the results in Table 5. As expected, compared to fine-tuning Qwen2.5-0.5B on its own selected data, fine-tuning on data selected by another LLM leads to a decline in TATA performance. This finding suggests that our TATA approach is base model-aware, emphasizing the principle of "teaching LLMs according to their aptitude." Interestingly, using data selected by LLMs within the same family (e.g., Qwen2.5-7B) yields more consistent performance compared to data selected by LLMs from a different family (LLaMA-3-8B). Complete results are in Appendix C.3.

## 6.3 EXPLORING REINFORCEMENT LEARNING

Recent advancements in RL (OpenAI, 2024; DeepSeek-AI et al., 2025) have demonstrated promising results in enhancing long CoT reasoning. To explore the role of RL in the spontaneous selection between CoT and TIR, we employ Direct Preference Optimization (DPO) to LLMs fine-tuned with our TATA framework (Rafailov et al., 2023) by constructing preference pairs based on the CoT and TIR scores of queries in the new candidate set $\mathcal{D}^* = \{(x_i, y_i^*, z_i^*)\}_{i=1}^N$. Detailed experimental setup and methodologies are provided in Appendix C.4. As shown in Table 6, DPO achieves results comparable to those of TATA. The complete results are provided in Table C.4. This suggests that the original data has already been effectively learned by the base LLM during the SFT stage, and applying additional DPO on the same dataset yields minor improvement. This observation aligns with LIMO (Ye et al., 2025), which argues that the capabilities of pretrained LLMs are latent, with both SFT and RL serving as different methods to elicit these inherent abilities.

Table 5: The best results (%) are **bold**, second-best underlined.

| Selected by | ID AVG | OOD AVG | AVG |
|---|---|---|---|
| TATA | **44.7** | **45.2** | **45.0** |
| LLaMA-3-8B | 43.8 | 44.2 | 44.1 |
| Qwen2.5-7B | 44.5 | 44.6 | 44.6 |

Table 6: DPO Results. Best results in **bold**.

| Model | Method | Acc | Token | # Code |
|---|---|---|---|---|
| LLaMA-3-8B | TATA | 61.5 | 371.7 | **1.32** |
| | +DPO | **61.6** | **365.4** | 1.34 |
| Qwen2.5Math-7B | TATA | **71.7** | **393.8** | **1.26** |
| | +DPO | **71.7** | 395.2 | 1.32 |

## 7 CONCLUSION

We propose TATA, a novel and effective framework for mathematical reasoning with LLMs that enables models to dynamically align their reasoning strategies, CoT or TIR, with their intrinsic strengths. By incorporating base-LLM-aware data selection during SFT, TATA tailors reasoning strategies to each model, empowering them to select an appropriate paradigm for inference autonomously. Extensive experiments demonstrate that TATA achieves superior or comparable performance across both in-domain and OOD benchmarks while significantly improving inference efficiency compared to method based on TIR alone. Moreover, our analysis underscores the importance of aptitude-aware data selection in unlocking the potential of LLMs to make autonomous and effective reasoning decisions, paving the way for further advancements in reasoning capabilities of LLMs.

REPRODUCIBILITY STATEMENT

All implementation details of our TATA framework are provided in Section 5.1 and Appendix B. Dataset curation procedures are described in Appendix B.1, while evaluation benchmarks are presented in Appendix B.2. The evaluation metrics are defined in Appendix B.3, and complete training details, including hyperparameters and model configurations, are given in Appendix B.4. We will release our code, training data, and models upon acceptance.

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

## A PRELIMINARIES

### A.1 REJECTION FINE-TUNING

For training LLMs, the original training datasets are often insufficient. To mitigate this issue, many studies adopt Rejection Fine-Tuning (RFT) (Yuan et al., 2023; Yu et al., 2023; Tong et al., 2024) to augment the original dataset, thereby increasing the training data size and improving model performance. RFT is a fine-tuning approach that uses synthesized data generated via rejection sampling (Yuan et al., 2023).

Suppose the original training set is $\mathcal{D}_{orig} = \{x_i, y_i\}_{i=1}^{N}$, consisting of $N$ data pairs $(x_i, y_i)$. The rejection sampling process works as follows: for each query $x_i$, a teacher model (e.g., GPT-4) generates $M$ responses, resulting in $\{x_i, y_i^j\}_{j=1}^{M}$, where $M$ is a predefined number (e.g., $M = 10$ in Yu et al. (2023)). This yields $N \cdot M$ response examples in total. A filtering process is then applied: if a response $y_i^j \neq y_i$, it is discarded. T he result is the augmented training set $\mathcal{D}_{aug} = \{x_i, y_i\}_{i=1}^{N}{}_{j=1}^{M_i}$, where $M_i \leq M$ represents the number of correct responses for query $x_i$. Notably, $M_i$ is often larger for simpler queries $x_i$, as these are more likely to produce correct responses.

RFT is widely employed for improving mathematical reasoning in LLMs (Yu et al., 2023; Tong et al., 2024; Xu et al., 2024c). Typically, the queries remain unchanged (Tong et al., 2024) or are altered in a controlled way (Yu et al., 2023). This is because the filtering stage of the rejection sampling process relies on the availability of ground-truth outputs.

## A.2 TIR Inference Pipeline

Tool-Integrated Reasoning (TIR) addresses mathematical problems by intertwining natural language reasoning with the execution of Python code. The process is initiated with gernerating a natural language reasoning step, denoted as $r_1$. When it is more advantageous to utilize programmatic tools, such as complex calculations, a Python code block, $a_1$, is created as guided by $r_1$. This code block is then run, and its result, $o_1$, is fed back into the model for further generation. This cycle is repeated until the maximal number of code blocks is reached or until the model concludes its answer within "\boxed{}." The entire reasoning path unfolds as $\tau = r_1 a_1 o_1 \ldots r_{n-1} a_{n-1} o_{n-1} r_n$, where $r_i$ is the $i$-th natural language reasoning step, $a_i$ denotes the corresponding Python code block, and $o_i$ represents the output from executing the code. The complete inference workflow is detailed in Algorithm 1 (from Gou et al. (2023)). From Algorithm 1, TIR usually requires multiple generations based on previous reasoning paths and outputs returned by Python interpreter, which is more computationally expensive than CoT. However, TIR can provide more precise calculation results than CoT.

---

**Algorithm 1** Inference of TIR

---

**Require:** problem $q$, model $\mathcal{G}$, prompt $p$, external tools $\mathcal{E}$, stop condition $Stop(\cdot)$, maximum iteration rounds $n$
1: $\tau_0 \leftarrow$ ""          $\triangleright$ Trajectory Initialization
2: **for** $i \leftarrow 1$ to $n$ **do**
3:     $r_i \sim \mathbb{P}_{\mathcal{G}}(\cdot | p \oplus q \oplus \tau_{i-1})$          $\triangleright$ Rationale Generation
4:     **if** $Stop(r_i)$ **then**          $\triangleright$ Stopping Criteria
5:        **return** $\tau_{i-1} \oplus r_i$
6:     **end if**
7:     $a_i \sim \mathbb{P}_{\mathcal{G}}(\cdot | p \oplus q \oplus \tau_{i-1} \oplus r_i)$          $\triangleright$ Program Generation
8:     $o_i \leftarrow \mathcal{E}(a_i)$          $\triangleright$ Tool Execution
9:     $\tau_i \leftarrow \tau_{i-1} \oplus r_i \oplus a_i \oplus o_i$          $\triangleright$ Trajectory Update
10: **end for**
11: **return** $\tau_n$

---

## A.3 Implicit Instruction Tuning

In-Context Learning (ICL) can be interpreted as a form of implicit instruction tuning, where the model is effectively "fine-tuned" using the given demonstrations in an implicit manner (Dai et al., 2022; Yang et al., 2023; Irie et al., 2022; Li et al., 2023). Let $\mathbf{X}_{\text{ins}}, \mathbf{X}_{\text{test}} \in \mathbb{R}^{d_{\text{in}}}$ represent the few-shot demonstration inputs and the test input, respectively. We define the attention query vector as $\boldsymbol{Q} = \mathbf{W}_Q \mathbf{X}_{\text{test}}^{\top}$, while the attention key and value vectors are given by $\boldsymbol{K} = \mathbf{W}_K [\mathbf{X}_{\text{ins}} \| \mathbf{X}_{\text{test}}]$ and $\boldsymbol{V} = \mathbf{W}_V [\mathbf{X}_{\text{ins}} \| \mathbf{X}_{\text{test}}]$, where $\|$ denotes concatenation. The projection matrices $\mathbf{W}_K, \mathbf{W}_V, \mathbf{W}_Q \in \mathbb{R}^{d_{\text{out}} \times d_{\text{in}}}$ are used to compute the attention queries, keys, and values. The self-attention mechanism for a single attention head in any given layer is formulated as follows:

$$\text{Attention}(\boldsymbol{K}, \boldsymbol{V}, \boldsymbol{Q}) =$$
$$\mathbf{W}_V [\mathbf{X}_{\text{ins}} \| \mathbf{X}_{\text{test}}] \text{Softmax} \left( \frac{\mathbf{W}_K [\mathbf{X}_{\text{ins}} \| \mathbf{X}_{\text{test}}]^{\top} \boldsymbol{Q}}{\sqrt{d_{\text{in}}}} \right).$$

Applying an approximation, this can be rewritten as:

$$\mathbf{W}_V [\mathbf{X}_{\text{ins}} \| \mathbf{X}_{\text{test}}] \left( \mathbf{W}_K [\mathbf{X}_{\text{ins}} \| \mathbf{X}_{\text{test}}] \right)^{\top} \boldsymbol{Q}.$$

By expanding this expression, we obtain:

$$\underbrace{\mathbf{W}_V \mathbf{X}_{\text{test}} (\mathbf{W}_K \mathbf{X}_{\text{test}})^{\top} \boldsymbol{Q}}_{\textit{Only test input.}} + \underbrace{\mathbf{W}_V \mathbf{X}_{\text{ins}} (\mathbf{W}_K \mathbf{X}_{\text{ins}})^{\top} \boldsymbol{Q}}_{\textit{Only demonstration samples.}}.$$

The whole approximation process can be given as follows:

$$
\begin{aligned}
&\text{Attention}(\boldsymbol{K}, \boldsymbol{V}, \boldsymbol{Q}) \\
&= \mathbf{W}_V[\mathbf{X}_{\text{ins}}\|\mathbf{X}_{\text{test}}]\text{Softmax}\left(\frac{\mathbf{W}_K[\mathbf{X}_{\text{ins}}\|\mathbf{X}_{\text{test}}]^\top \boldsymbol{Q}}{\sqrt{d_{\text{in}}}}\right) \\
&\approx \mathbf{W}_V[\mathbf{X}_{\text{ins}}\|\mathbf{X}_{\text{test}}]\left(\mathbf{W}_K[\mathbf{X}_{\text{ins}}\|\mathbf{X}_{\text{test}}]\right)^\top \boldsymbol{Q} \\
&= \underbrace{\mathbf{W}_V\mathbf{X}_{\text{test}}(\mathbf{W}_K\mathbf{X}_{\text{test}})^\top \boldsymbol{Q}}_{\text{Only test input.}} + \underbrace{\mathbf{W}_V\mathbf{X}_{\text{ins}}(\mathbf{W}_K\mathbf{X}_{\text{ins}})^\top \boldsymbol{Q}}_{\text{Only instruction sample.}} \\
&= [\underbrace{\mathbf{W}_V\mathbf{X}_{\text{test}}(\mathbf{W}_K\mathbf{X}_{\text{test}})^\top}_{\text{Only test input.}} + \underbrace{\mathbf{W}_V\mathbf{X}_{\text{ins}}(\mathbf{W}_K\mathbf{X}_{\text{ins}})^\top}_{\text{Only instruction sample.}}]\boldsymbol{Q},
\end{aligned}
$$

where the constant $\sqrt{d_{\text{in}}}$ acts as a scaling factor. The first term, $\mathbf{W}_V\mathbf{X}_{\text{test}}(\mathbf{W}_K\mathbf{X}_{\text{test}})^\top$, corresponds to a zero-shot learning scenario where no demonstration samples are involved, and only the test input is considered. Meanwhile, the second term, $\mathbf{W}_V\mathbf{X}_{\text{ins}}(\mathbf{W}_K\mathbf{X}_{\text{ins}})^\top$, can be interpreted as an implicit adjustment to the model parameters. This adjustment is achieved through the meta-gradient mechanism (Dai et al., 2022; Yang et al., 2023; Irie et al., 2022), meaning the few-shot examples influence the model as if performing implicit instruction tuning.

# B EXPERIMENTAL SETUP

## B.1 TATA IMPLEMENTATION DETAILS

In this appendix, we give the implementation details of our TATA framework.

**Data Construction** For the original training set, denoted as $\mathcal{D}_{\text{orig}} = \{(x_i, y_i)\}_{i=1}^N$, we utilize the training sets of GSM8K (Cobbe et al., 2021) and MATH (Hendrycks et al., 2021). The GSM8K training set comprises 7,473 examples, while the MATH training set includes 7,500 examples. For simplicity, we directly adopt the DART-MATH-Hard dataset (Tong et al., 2024) as our $\mathcal{D}_{\text{aug}}$. DART-MATH-Hard, which is an augmented dataset derived from the GSM8K and MATH training sets through rejection sampling, contains approximately 0.6M examples in total. Notably, the number of responses varies across different training queries. To convert CoT solutions into TIR format, we use `GPT-4o-2024-08-06` with a carefully designed prompt, as described in Table 7. While most CoT solutions are successfully transformed into TIR format, we observe some anomalies. For instance, some rewritten TIRs fail to conclude with a final answer, while some TIRs produce code with syntax errors. To address these issues, we filter out ill-formed TIRs using rule-based matching. After filtering, we obtain a candidate dataset containing approximately 483K examples.

**Anchor Construction** For the embedding, we use `text-embedding-ada-002` to encode all queries in our candidate set $\mathcal{D}$ into 1,536-dimensional vectors. We then cluster these representations by K-means algorithm. We set the number of clusters to be 100 for both GSM8K and MATH (cluster separately). That is to say, the size of the anchor set is $A = 100$.

**Contribution Quantification** To compute the CoT and TIR scores, we use a new candidate set, denoted as $\mathcal{D}^* = \{(x_i, y_i^*, z_i^*)\}_{i=1}^N$. This new candidate set is constructed by randomly selecting one pair of CoT and TIR solutions for each training query from the original candidate set, thereby reducing computational costs. The CoT score is then simplified to:

$$
S_{\text{CoT}}^k = \frac{1}{A}\sum_{i=1}^{A}\mathbb{I}\big(a_i, \mathcal{G}(\cdot \mid \underbrace{x_k, y^*}_{\text{1-shot prompt}}, q_i)\big),
$$

A similar formulation is used for the TIR score.

**Data Selection** The distributions of $(S_{\text{CoT}}^k - S_{\text{TIR}}^k)$ on GSM8K and MATH reveal distinct patterns (see Section 6.1 and Appendix C.2): all base LLMs demonstrate a tendency to rely more on CoT for GSM8K queries, while preferring TIR for MATH queries. As a result, it is reasonable to select

---

**Rewriting Prompt Template**

You are a helpful mathematical assistant. A problem will be presented after "Problem:", followed by a reference solution after "Original Solution:". Your task is to rewrite the original solution. During rewriting, you tend to leverage Python (sympy is preferred) to facilitate solving the problem with step-by-step reasoning, especially for calculation and simplification. The specific requirements are as follows:

1. Analyze the problem and write functions to solve it, ensuring that the functions do not require any arguments.

2. Present the final result in LaTeX using a $\boxed{\text{ANS}}$ without any units.

3. Utilize the 'pi' symbol and 'Rational' from Sympy for $\pi$ and fractions, and simplify all fractions and square roots without converting them to decimal values.

4. Avoid using sentences like "Reasoning step in natural language:", "Reasoning in Python codes:", and other similar phrases.

5. Combine multiple calculation steps with Python code blocks where appropriate, avoiding unnecessary separate blocks. Limit the number of Python code blocks to fewer than 5 and use them wisely.

6. The new solution format should be as follows:

"Reasoning step 1 in natural language without specific calculations
```python
Python code block 1 for calculation and simplification, please print out the final output using `print`
```
```output
The output for code block 1
```
......
Reasoning step N in natural language without specific calculations
```python
Python code block N for calculation and simplification, please print out the final output using `print`
```
```output
The output for code block N
```
Conclude the final answer."

Problem: {problem}

Original Solution: {raw_answer}

New Solution:

---

Table 7: The prompt for transforming CoT to TIR.

different decision functions, $\mathcal{H}$, for GSM8K and MATH. Specifically, for GSM8K, the dataset for supervised fine-tuning ($D_{\text{SFT}}$) is defined as:

$$D_{\text{SFT}} = \bigcup_{k=1}^{N} \{(x_k, y_k^j)\}_{j=1}^{M_k} \cup \bigcup_{k \in A} \{(x_k, z_k^j)\}_{j=1}^{M_k},$$

where the index set $A = \{k : S_{\text{CoT}}^k - S_{\text{TIR}}^k < \text{quantile}_1\}$.

For MATH, $D_{\text{SFT}}$ is defined as:

$$D_{\text{SFT}} = \bigcup_{k=1}^{N} \{(x_k, z_k^j)\}_{j=1}^{M_k} \cup \bigcup_{k \in B} \{(x_k, y_k^j)\}_{j=1}^{M_k},$$

where the index set $B = \{k : S_{\text{CoT}}^k - S_{\text{TIR}}^k > \text{quantile}_2\}$.

The thresholds quantile$_1$ and quantile$_2$ are determined through grid search. Notably, the performance of TATA is not sensitive to these quantiles (see Section 5.3 and Table 10). Additionally, we explored alternative decision functions $\mathcal{H}$ in our ablation study, with further details provided in Section 5.3 and Appendix C.1.

| Model | Quantiles | Metric | In-Domain | | | Out-of-Domain | | | | | AVG |
| --- | --- | --- | --- | --- | --- | --- | --- | --- | --- | --- | --- |
| | | | GSM8K | MATH | ID AVG | MAWPS | SVAMP | College | Olympiad | OOD AVG | |
| Qwen2.5-0.5B | 50, 60 | Acc | 52.2 | 37.2 | 44.7 | 86.4 | 55.7 | 27.5 | 9.9 | 44.9 | 44.8 |
| | | Token | 313.5 | 503.1 | 408.3 | 224.3 | 304.7 | 496.1 | 748.2 | 443.3 | 431.7 |
| | | # Code | 0.2 | 2.62 | 1.41 | 0.63 | 0.32 | 2.85 | 3.03 | 1.71 | 1.61 |
| | 40, 60 | Acc | 53.5 | 36.4 | **45.0** | 85.9 | 57.9 | 26.4 | 8.4 | 44.7 | 44.8 |
| | | Token | 307.2 | 504.2 | 405.7 | 217.7 | 290.6 | 486.8 | 715.2 | 427.6 | 420.3 |
| | | # Code | 0.24 | 2.5 | 1.37 | 0.56 | 0.3 | 2.7 | 2.84 | 1.6 | 1.52 |
| | 30, 60 | Acc | 53.1 | 37.0 | **45.0** | 86.2 | 56.3 | 26.7 | 10.2 | 44.8 | 44.9 |
| | | Token | 312.7 | 507.5 | 410.1 | 218.6 | 298.1 | 482.4 | 720.6 | 429.9 | 423.3 |
| | | # Code | 0.21 | 2.49 | 1.35 | 0.49 | 0.29 | 2.73 | 2.81 | 1.58 | 1.50 |
| | 30, 65* | Acc | 52.8 | 36.6 | 44.7 | 85.9 | 59.4 | 26.9 | 8.6 | **45.2** | **45.0** |
| | | Token | 309.7 | 508.7 | 409.2 | 217.3 | 292.9 | 500.9 | 743.0 | 438.5 | 428.8 |
| | | # Code | 0.19 | 2.63 | 1.41 | 0.52 | 0.33 | 2.82 | 3.06 | 1.68 | 1.59 |
| | 30, 70 | Acc | 52.2 | 37.1 | 44.7 | 86.4 | 55.7 | 27.6 | 9.9 | 44.9 | 44.8 |
| | | Token | 313.5 | 503.1 | 408.3 | 224.3 | 304.7 | 496.1 | 748.2 | 443.3 | 431.7 |
| | | # Code | 0.2 | 2.62 | 1.41 | 0.63 | 0.32 | 2.85 | 3.03 | 1.71 | 1.61 |

Table 8: Performance across different quantiles using Qwen2.5-0.5B. The best accuracies within each group are shown in **bold**. The three metrics, "Acc", "Token", and "# Code" represent the average accuracy, total tokens per generation, and number of code executions. "Acc" is reported in %. "ID AVG", "OOD AVG", and "AVG" denote the averages of these metrics across in-domain, out-of-domain, and all six benchmarks. The two numbers in the "Quantiles" are the quantile of GSM8K and MATH, respectively. * denote our chosen quantiles.

## B.2 EVALUATION BENCHMARKS

We give a brief introduction of evaluated benchmarks mentioned in Section 5.1.

- GSM8K (Cobbe et al., 2021) is a grade-school math benchmark, consisting of 7,473 training examples and 1,319 test examples. It is available at this link, and under MIT License.
- MATH (Hendrycks et al., 2021) is a competition-level math dataset, including 5,000 test examples and 7,500 training examples. It is available at this link, and under MIT License.
- MAWPS (Koncel-Kedziorski et al., 2016) is a benchmark of math word problems (MWPs), incorporating 238 test examples. It is under MIT License and can be found at https://github.com/LYH-YF/MWPToolkit.
- SVAMP (Patel et al., 2021) includes 1,000 simple MWPs, which is available at https://github.com/LYH-YF/MWPToolkit. It is under MIT License.
- CollegeMath (Tang et al., 2024): This dataset comprises 2818 college-grade mathematical questions sourced from 9 different textbooks, covering 7 fields including linear algebra and differential equations. It is designed to evaluate generalization in intricate mathematical reasoning across various domains. It is available at this link.

- OlympiadBench-Math (He et al., 2024): This collection comprises 675 high-level Olympiad mathematical problems selected from various competitions and represents a text-only English fraction of OlympiadBench. It is available at this link.

## B.3 EVALUATION METRICS

In addition to evaluating accuracy across the six benchmarks mentioned in Section 5.1, we also assess the computational costs associated with interacting with external Python interpreters. As described in Algorithm 1, TIR involves multiple interactions with Python interpreters. The associated time costs can be divided into two categories: the time required to execute Python code blocks and the increased generation costs caused by progressively longer input sequences. The first type of time cost is reflected in the number of interactions with Python interpreters, i.e., the number of code executions. The second type can be approximated by the number of generated tokens, which includes both input and output tokens. Since the number of generations is equivalent to the number of code executions, we use the average total tokens per generation to evaluate this cost. Naturally, TIR incurs a higher number of generated tokens due to multiple generations with progressively longer contexts.

## B.4 SFT AND EVALUATION SETUP

**SFT Setup**    In our experiments, we utilize various base LLMs, including general-purpose models (e.g., LLaMA-3-8B (AI@Meta, 2024)) and math-specialized models (e.g., Qwen2.5-Math (Yang et al., 2024b)). The details of these base LLMs are outlined below:

- **Llama-3** (AI@Meta, 2024): LLaMA 3 Community License. We use Llama-3-8B as the base LLM in our experiments.

- **Qwen2.5** (Yang et al., 2024a): Qwen2.5 series are developed with dedication to math and coding. We used 0.5B, 1.5B, 3B, and, 7B models. They are licensed under Apache 2.0.

- **Qwen2.5-Math** (Yang et al., 2024b): Qwen2.5-Math is a series of specialized math language models built upon the Qwen2.5 LLMs. We use 3B and 7B variants. They are under the same license as the Qwen2.5 series.

We set the maximum input length for all base models to be 4,096. During SFT, we employ the Adam optimizer with a learning rate of $2 \times 10^{-5}$ and set batch size to 64, conducting training over three epochs. Unlike Beeching et al. (2024); Yang et al. (2024b), we use the same training prompt for both CoT and TIR. The prompt is provided in Table 9.

| **Training and Inference Prompt Template** |
| --- |
| Below is an instruction that describes a task. Write a response that appropriately completes the request. |
| **### Instruction:** 
 {instruction} |
| **### Response:** |

Table 9: Training prompt for base LLMs.

**Evaluation Setup**    For evaluation, we adopt the same prompt used during SFT, as recommended by Tong et al. (2024). For TIR inference, please refer to Algorithm 1, where the maximum number of interactions is set to $n = 6$. CoT inference can be viewed as a special case of Algorithm 1 with $n = 1$.

# C MORE FINE-GRAINED RESULTS

## C.1 ABLATION STUDY

As detailed in Appendix B, we use different decision function $\mathcal{H}$ for GSM8K and MATH. Specifically, for GSM8K, the dataset for supervised fine-tuning ($D_{\text{SFT}}$) is defined as:

$$D_{\text{SFT}} = \bigcup_{k=1}^{N} \{(x_k, y_k^j)\}_{j=1}^{M_k} \cup \bigcup_{k \in A} \{(x_k, z_k^j)\}_{j=1}^{M_k},$$

where the index set $A = \{k : S_{\text{CoT}}^k - S_{\text{TIR}}^k < \text{quantile}_1\}$.

For MATH, $D_{\text{SFT}}$ is defined as:

$$D_{\text{SFT}} = \bigcup_{k=1}^{N} \{(x_k, z_k^j)\}_{j=1}^{M_k} \cup \bigcup_{k \in B} \{(x_k, y_k^j)\}_{j=1}^{M_k},$$

where the index set $B = \{k : S_{\text{CoT}}^k - S_{\text{TIR}}^k > \text{quantile}_2\}$. We consider this as the default choice of our TATA (i.e., TATA in Table 10).

We present the results of the $\mathcal{H}$ ablation study in Table 10. The variants of $\mathcal{H}$ evaluated are described as follows:

**Random** The key difference between "Random" and "TATA" lies in the selection of the index sets $A$ and $B$. In the "Random" variant, we randomly select the index sets $A$ and $B$ while ensuring that $|A|$ and $|B|$ match those in the default TATA configuration. It is important to note that this is not purely a random selection, the number of queries using TIR or CoT is still determined by the default settings of TATA, making "Random" a strong baseline.

**CoT + TIR** In this variant, we include all CoT and TIR solutions in $D_{\text{SFT}}$, doubling the number of training examples compared to using only CoT or TIR individually. Formally, the dataset is defined as:

$$D_{\text{SFT}} = \bigcup_{k=1}^{N} \{(x_k, y_k^j)\}_{j=1}^{M_k} \cup \bigcup_{k=1}^{N} \{(x_k, z_k^j)\}_{j=1}^{M_k}.$$

**TATA$^-$** The TATA$^-$ variant differs from the original TATA in that it uses a single quantile for selection. The dataset is formally defined as:

$$D_{\text{SFT}} = \bigcup_{k \in A} \{(x_k, y_k^j)\}_{j=1}^{M_k} \cup \bigcup_{k \in B} \{(x_k, z_k^j)\}_{j=1}^{M_k},$$

where the index set $A = \{k : S_{\text{CoT}}^k - S_{\text{TIR}}^k > \text{quantile}\}$, and $B = A^c$. In this setup, each query in the candidate set $\mathcal{D}^* = \{(x_i, y_i^*, z_i^*)\}_{i=1}^{N}$ includes either CoT or TIR solutions but not both.

From Table 10, the selection function $\mathcal{H}$ in our TATA gains the best results.

## C.2 ANALYSIS OF COT SCORES AND TIR SCORES

In Section 6.1, we presented representative results analyzing CoT and TIR scores. Here, we further provide the distributions of $S_{\text{CoT}}^k$, $S_{\text{TIR}}^k$, and $(S_{\text{CoT}}^k - S_{\text{TIR}}^k)$ for various base LLMs in Figures 4, 5, 6, 7, 8, 9, and 10. From these figures, we have the following observations: 1. Different base LLMs exhibit varying tendencies towards CoT or TIR responding to the same candidate set queries. 2. Math-specialized LLMs (e.g., Qwen2.5Math) demonstrate higher CoT and TIR scores compared to their general-purpose counterparts (e.g., Qwen2.5). This may be attributed to the inclusion of similar CoT and TIR data in their pretraining process. 3. Notably, Qwen2.5Math-7B achieves TIR scores approaching 0.8 accuracy on the MATH anchor set using only a 1-shot prompt from the candidate set, as shown in Figure 10 (middle). This suggests the potential for anchor set contamination (Xu et al., 2024a).

| Model | Method | Metric | In-Domain | | | Out-of-Domain | | | | | AVG |
|---|---|---|---|---|---|---|---|---|---|---|---|
| | | | GSM8K | MATH | ID AVG | MAWPS | SVAMP | College | Olympiad | OOD AVG | |
| LLaMA-3-8B | CoT | Acc | **84.7** | 46.5 | 65.6 | 91.6 | 81.6 | 30.2 | 13.3 | 54.2 | 58.0 |
| | | Token | 246.4 | 471.0 | 358.7 | 173.3 | 236.8 | 511.7 | 676.7 | 399.6 | 386.0 |
| | | # Code | 0.0 | 0.0 | 0.0 | 0.0 | 0.0 | 0.0 | 0.0 | 0.0 | 0.0 |
| | TIR | Acc | 81.7 | 56.2 | 69.0 | 87.8 | 77.8 | 30.5 | **21.9** | 54.5 | 59.3 |
| | | Token | 299.0 | 457.5 | 378.2 | 240.9 | 269.1 | 437.9 | 650.8 | 399.7 | 392.5 |
| | | # Code | 2.96 | 2.51 | 2.74 | 2.42 | 2.64 | 2.69 | 2.76 | 2.63 | 2.66 |
| | Random | Acc | 83.1 | **56.4** | **69.8** | 91.8 | 81.3 | 31.3 | 21.8 | 56.6 | 61.0 |
| | | Token | 271.6 | 472.0 | 371.8 | 203.7 | 251.0 | 453.4 | 695.5 | 400.9 | 391.2 |
| | | # Code | 0.21 | 2.35 | 1.28 | 0.36 | 0.33 | 2.44 | 2.83 | 1.49 | 1.42 |
| | CoT + TIR | Acc | 83.1 | 48.4 | 65.8 | 91.2 | 78.7 | 30.8 | 16.7 | 54.4 | 58.2 |
| | | Token | 278.0 | 497.4 | 387.7 | 208.6 | 281.2 | 507.3 | 707.3 | 421.1 | 410.0 |
| | | # Code | 0.83 | 0.51 | 0.67 | 0.68 | 0.95 | 0.51 | 1.09 | 0.81 | 0.76 |
| | TATA⁻ | Acc | 83.1 | 54.7 | 68.9 | 91.2 | 80.6 | 31.9 | 19.6 | 55.8 | 60.2 |
| | | Token | 285.4 | 472.1 | 378.8 | 226.7 | 253.9 | 474.3 | 692.2 | 411.8 | 400.8 |
| | | # Code | 1.4 | 2.31 | 1.86 | 1.23 | 1.2 | 2.34 | 2.49 | 1.81 | 1.83 |
| | TATA | Acc | 84.0 | 55.1 | 69.6 | **91.8** | **82.7** | **34.2** | 21.5 | **57.6** | **61.5** |
| | | Token | 248.2 | 461.1 | 354.6 | 191.1 | 222.5 | 449.5 | 657.7 | 380.2 | 371.7 |
| | | # Code | 0.12 | 2.33 | 1.23 | 0.27 | 0.21 | 2.39 | 2.6 | 1.37 | 1.32 |

Table 10: Ablation Study using LLaMA-3-8B. The best accuracies within each group are shown in **bold**. The three metrics, "Acc", "Token", and "# Code" represent the average accuracy, total tokens per generation, and number of code executions. "Acc" is reported in %. "ID AVG", "OOD AVG", and "AVG" denote the averages of these metrics across in-domain, out-of-domain, and all six benchmarks.

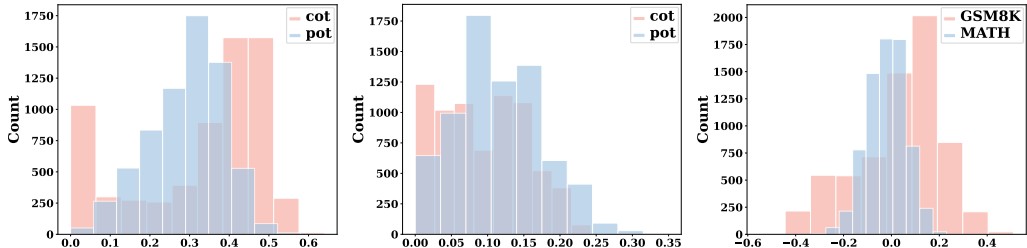

Figure 4: The distribution of $S_{\text{CoT}}^k$ (left), $S_{\text{TIR}}^k$ (middle), and $(S_{\text{CoT}}^k - S_{\text{TIR}}^k)$ (right) for LLaMA-3-8B.

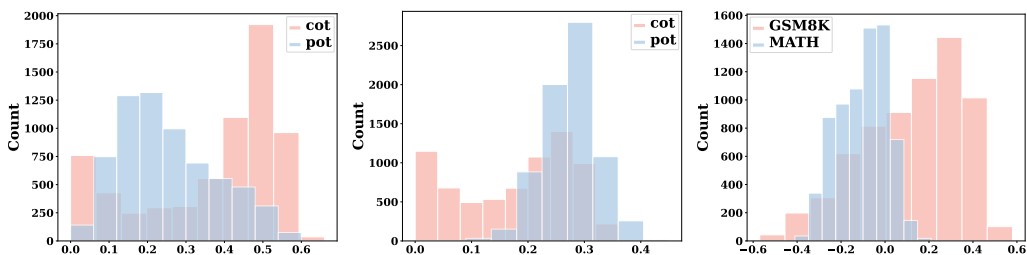

Figure 5: The distribution of $S_{\text{CoT}}^k$ (left), $S_{\text{TIR}}^k$ (middle), and $(S_{\text{CoT}}^k - S_{\text{TIR}}^k)$ (right) for Qwen2.5-0.5B.

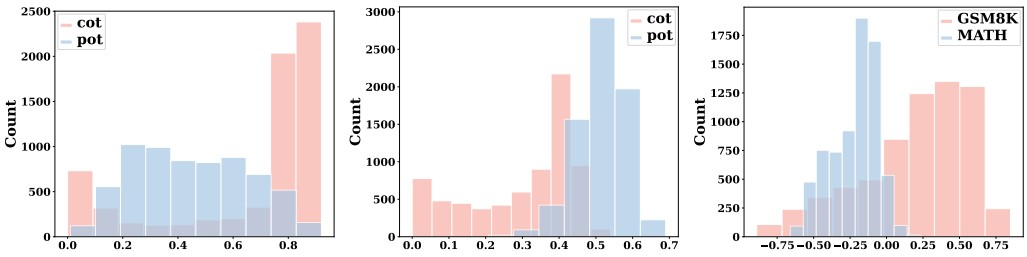

Figure 6: The distribution of $S_{\text{CoT}}^k$ (left), $S_{\text{TIR}}^k$ (middle), and $(S_{\text{CoT}}^k - S_{\text{TIR}}^k)$ (right) for Qwen2.5-1.5B.

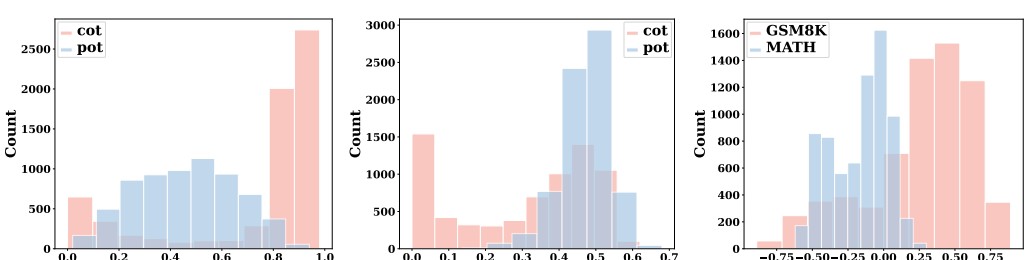

Figure 7: The distribution of $S_{\text{CoT}}^k$ (left), $S_{\text{TIR}}^k$ (middle), and $(S_{\text{CoT}}^k - S_{\text{TIR}}^k)$ (right) for Qwen2.5-3B.

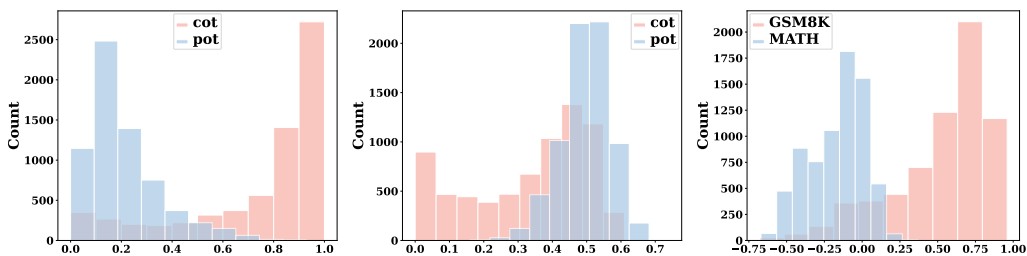

Figure 8: The distribution of $S_{\text{CoT}}^k$ (left), $S_{\text{TIR}}^k$ (middle), and $(S_{\text{CoT}}^k - S_{\text{TIR}}^k)$ (right) for Qwen2.5-7B.

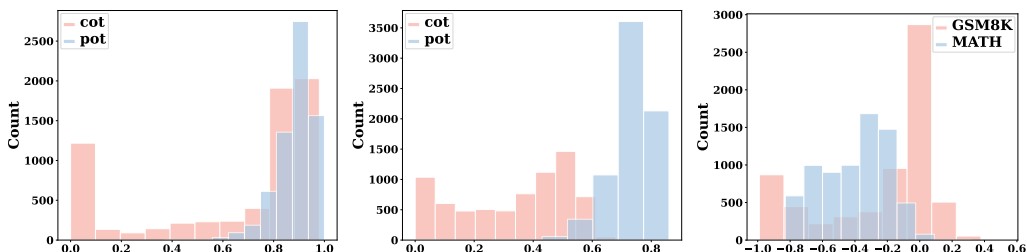

Figure 9: The distribution of $S_{\text{CoT}}^k$ (left), $S_{\text{TIR}}^k$ (middle), and $(S_{\text{CoT}}^k - S_{\text{TIR}}^k)$ (right) for Qwen2.5Math-1.5B.

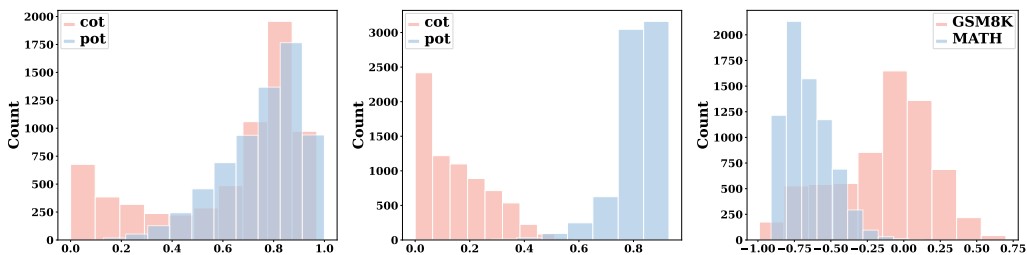

Figure 10: The distribution of $S_{\text{CoT}}^k$ (left), $S_{\text{TIR}}^k$ (middle), and $(S_{\text{CoT}}^k - S_{\text{TIR}}^k)$ (right) for Qwen2.5Math-7B.

## C.3 TRANSFERABILITY RESULTS

The complete results of transferability results are given in Table 11.

| Model | Select By | Metric | In-Domain | | | Out-of-Domain | | | | | AVG |
|---|---|---|---|---|---|---|---|---|---|---|---|
| | | | GSM8K | MATH | ID AVG | MAWPS | SVAMP | College | Olympiad | OOD AVG | |
| Qwen2.5-0.5B | Qwen2.5-0.5B | Acc | **52.8** | 36.6 | **44.7** | 85.9 | **59.4** | **26.9** | **8.6** | 45.2 | **45.0** |
| | | Token | 309.7 | 508.7 | 409.2 | 217.3 | 292.9 | 500.9 | 743.0 | 438.5 | 428.8 |
| | | # Code | 0.19 | 2.63 | 1.41 | 0.52 | 0.33 | 2.82 | 3.06 | 1.68 | 1.59 |
| | LLaMA-3-8B | Acc | 51.3 | 36.3 | 43.8 | 86.2 | 55.9 | 26.5 | 8.1 | 44.2 | 44.1 |
| | | Token | 318.2 | 507.7 | 413.0 | 216.9 | 298.9 | 485.4 | 732.8 | 433.5 | 426.6 |
| | | # Code | 0.28 | 2.49 | 1.39 | 0.52 | 0.52 | 2.45 | 2.73 | 1.56 | 1.5 |
| | Qwen2.5-7B | Acc | 52.2 | 36.8 | 44.5 | **86.7** | 57.6 | 26.7 | 7.4 | 44.6 | **44.6** |
| | | Token | 312.5 | 499.4 | 406.0 | 228.6 | 308.2 | 489.3 | 744.5 | 442.6 | 430.4 |
| | | # Code | 0.4 | 2.53 | 1.46 | 0.85 | 0.68 | 2.75 | 2.94 | 1.81 | 1.69 |

Table 11: Detailed results of transferability experiments using Qwen2.5-0.5B. The best accuracies within each group are shown in **bold**. The three metrics, "Acc", "Token", and "# Code" represent the average accuracy, total tokens per generation, and number of code executions. "Acc" is reported in %. "ID AVG", "OOD AVG", and "AVG" denote the averages of these metrics across in-domain, out-of-domain, and all six benchmarks.

## C.4 DPO RESULTS

The detailed settings of DPO are as follows:

**Preference Data Construction** The construction of the preference dataset used in DPO is guided by CoT and TIR scores, following a similar approach to the construction of $\mathcal{D}_{\text{SFT}}$. Specifically, two separate quantiles are used to select preference pairs for the GSM8K and MATH datasets. The preference dataset, $\mathcal{D}_{\text{pre}}$, is selected from the newly defined candidate set, $\mathcal{D}^* = \{(x_i, y_i^*, z_i^*)\}_{i=1}^N$, and is formally defined as:

$$\mathcal{D}_{\text{pre}} = \{(x_k, c_k, r_k)\}_{k \in A},$$

where $c_k$ is the **c**hosen (preferred) response for the query $x_k$, and $r_k$ is the **r**ejected response.

The index set $A$ is defined as:

$$A = \{k : S_{\text{TIR}}^k - S_{\text{CoT}}^k < \text{quantile}_1^{'} \quad \text{or}$$
$$S_{\text{CoT}}^k - S_{\text{TIR}}^k > \text{quantile}_2^{'}\},$$

where $\text{quantile}_1^{'}$ and $\text{quantile}_2^{'}$ are two quantiles optimized via grid search.

The rules for determining $c_k$ (chosen response) and $r_k$ (rejected response) are as follows:

$$c_k = \begin{cases} y_k & \text{if } S_{\text{CoT}}^k - S_{\text{TIR}}^k > \text{quantile}_2^{'}, \\ z_k & \text{if } S_{\text{TIR}}^k - S_{\text{CoT}}^k < \text{quantile}_1^{'}, \end{cases}$$

and

$$r_k = \begin{cases} y_k & \text{if } S_{\text{TIR}}^k - S_{\text{CoT}}^k < \text{quantile}_1^{'}, \\ z_k & \text{if } S_{\text{CoT}}^k - S_{\text{TIR}}^k > \text{quantile}_2^{'}. \end{cases}$$

This preference selection process ensures that the dataset $\mathcal{D}_{\text{pre}}$ contains meaningful comparisons between CoT and TIR responses based on their relative scores.

**DPO Hyperparameters** We utilize OpenRLHF (Hu et al., 2024) to implement DPO. The maximum token length is set to 4,096, consistent with the SFT stage. The training process adopts a learning rate of $5 \times 10^{-7}$, a batch size of 256, and runs for one epoch. We use LLaMA-3-8B and Qwen2.5Math-7B, fine-tuned with TATA, as the starting point for DPO.

The complete results are presented in Table 12. As shown, DPO achieves comparable results with LLMs fine-tuned with TATA.

| Model | Method | Metric | In-Domain | | | Out-of-Domain | | | | | AVG |
|-------|--------|--------|-------|------|--------|-------|-------|---------|----------|---------|------|
| | | | GSM8K | MATH | ID AVG | MAWPS | SVAMP | College | Olympiad | OOD AVG | |
| LLaMA-3-8B | CoT | Acc | **84.7** | 46.5 | 65.6 | 91.6 | 81.6 | 30.2 | 13.3 | 54.2 | 58.0 |
| | | Token | 246.4 | 471.0 | 358.7 | 173.3 | 236.8 | 511.7 | 676.7 | 399.6 | 386.0 |
| | | # Code | 0.0 | 0.0 | 0.0 | 0.0 | 0.0 | 0.0 | 0.0 | 0.0 | 0.0 |
| | TIR | Acc | 81.7 | **56.2** | 69.0 | 87.8 | 77.8 | 30.5 | **21.9** | 54.5 | 59.3 |
| | | Token | 299.0 | 457.5 | 378.2 | 240.9 | 269.1 | 437.9 | 650.8 | 399.7 | 392.5 |
| | | # Code | 2.96 | 2.51 | 2.74 | 2.42 | 2.64 | 2.69 | 2.76 | 2.63 | 2.66 |
| | TATA | Acc | 84.0 | 55.1 | **69.6** | **91.8** | **82.7** | **34.2** | 21.5 | **57.6** | 61.5 |
| | | Token | 248.2 | 461.1 | 354.6 | 191.1 | 222.5 | 449.5 | 657.7 | 380.2 | 371.7 |
| | | # Code | 0.12 | 2.33 | 1.23 | 0.27 | 0.21 | 2.39 | 2.6 | 1.37 | 1.32 |
| | +DPO | Acc | 84.0 | 55.2 | **69.6** | **91.8** | **82.7** | 34.0 | 21.8 | **57.6** | **61.6** |
| | | Token | 250.8 | **453.6** | **352.2** | **185.0** | **219.1** | 435.9 | 647.9 | 372.0 | **365.4** |
| | | # Code | 0.14 | 2.38 | 1.26 | 0.25 | 0.17 | 2.42 | 2.7 | 1.38 | 1.34 |
| Qwen2.5Math-7B | CoT | Acc | **91.0** | 61.5 | 76.2 | 94.8 | 87.9 | 45.7 | 23.9 | 63.1 | 67.5 |
| | | Token | **254.7** | 470.6 | 362.6 | **177.0** | 223.5 | 484.1 | 669.2 | 388.5 | 379.9 |
| | | # Code | 0.0 | 0.0 | 0.0 | 0.0 | 0.0 | 0.0 | 0.01 | 0.0 | 0.0 |
| | TIR | Acc | 88.9 | **73.6** | 81.2 | **95.4** | **89.4** | 47.1 | 35.3 | 66.8 | 71.6 |
| | | Token | 311.8 | 490.9 | 401.4 | 261.2 | 272.2 | **456.8** | 713.7 | 426.0 | 417.8 |
| | | # Code | 3.04 | 2.56 | 2.8 | 2.58 | 2.51 | 2.65 | 2.75 | 2.62 | 2.68 |
| | TATA | Acc | 89.8 | 73.0 | **81.4** | 95.2 | 88.1 | **48.3** | **35.9** | **66.9** | **71.7** |
| | | Token | 264.7 | 487.2 | 376.0 | 193.7 | 229.7 | 476.9 | 710.6 | 402.7 | 393.8 |
| | | # Code | 0.25 | 2.14 | 1.2 | 0.33 | 0.24 | 2.02 | 2.59 | 1.3 | 1.26 |
| | +DPO | Acc | 89.8 | 73.1 | **81.4** | 95.2 | 88.1 | **48.4** | 35.4 | 66.8 | **71.7** |
| | | Token | 267.0 | **487.2** | 377.1 | 193.8 | 229.4 | 474.8 | 718.9 | 404.2 | 395.2 |
| | | # Code | 0.3 | 2.18 | 1.24 | 0.39 | 0.27 | 2.08 | 2.67 | 1.35 | 1.32 |

Table 12: Detailed DPO results. The best accuracies within each group are shown in **bold**. The three metrics, "Acc", "Token", and "# Code" represent the average accuracy, total tokens per generation, and number of code executions. "Acc" is reported in %. "ID AVG", "OOD AVG", and "AVG" denote the averages of these metrics across in-domain, out-of-domain, and all six benchmarks.

# D    THE LLM USAGE DECLARATION

In this work, we employ **GPT-4o** to transform CoT answers into the TIR format, as described in Section 4. As one of our baselines, we also use GPT-4o for SFT data selection, denoted as "GPT-Select" in Table 1. In addition, we incorporate several base models for our SFT experiments. Finally, we utilize **GPT-5** to assist in refining our writing.

