# OpenReview forum: "Teaching LLMs According to Their Aptitude: Adaptive Switching Between CoT and TIR for Mathematical Problem Solving"
_ICLR.cc/2026/Conference — ICLR 2026 Conference Withdrawn Submission_

### Official Review · Reviewer_WnPk · 2025-10-16

**Soundness:** 2
**Presentation:** 3
**Contribution:** 2
**Rating:** 2
**Confidence:** 4

**Summary:**

This paper argues that CoT and TIR play different roles and have different features in the reasoning process of large language models.

Based on this assumption, the authors propose a new method that decides whether to use CoT or TIR for reasoning before the model starts to think. Specifically, for each question in the training set, the authors create both CoT and TIR data and compute the corresponding CoT score and TIR score to train the model.

They evaluate their method on several mathematical tasks and achieve improvements on downstream tasks. In addition, the authors further analyze the algorithm, examining the effect of each module and the additional gains brought by reinforcement learning.

**Strengths:**

+ Achieve better performance than other baseline approaches.

+ Experiments were conducted on multiple models (including Llama and Qwen), and the method proved to be effective on all of them.

+ The method is relatively simple and can be applied more easily.

**Weaknesses:**

+ The paper does not verify the respective advantages of CoT and TIR. It should discuss which types of questions are more suitable for CoT and which are more suitable for TIR, otherwise the paper may lack persuasiveness.

+ This method is similar to multi-agent cooperation. The paper only discusses two reasoning forms, CoT and TIR. When more reasoning forms such as ToT and MCTS are introduced, it is unclear whether the algorithm will still be effective. In multi-agent cooperation studies, multiple agents are usually involved, while this paper only uses two reasoning forms, which limits its overall contribution.

+ The evaluation process of the paper is not rigorous. The settings of the In-Domain and Out-of-Domain tests are unreasonable, as both are mathematical reasoning tasks and should be considered In-Domain. In addition, in the ablation study, removing certain modules causes only very limited performance degradation, which can be regarded as normal performance fluctuation. This makes it difficult to prove that each component of the algorithm is truly effective.

+ The potential of this algorithm is limited, as it cannot be continuously improved through reinforcement learning, and its practical value is therefore restricted.

**Questions:**

+ In line 015, the phrase “CoT for generalizability” is mentioned. Please further explain why CoT is beneficial for generalizability.

+ What are the advantages of CoT and TIR, and why does combining them lead to better performance?

+ In Sec. 4.3 Contribution Quantification, why is one-shot used? Would using more examples change the distribution of the scores? This is because some models might require more examples to learn TIR.

+ In Table 1, both In-Domain and Out-of-Domain tasks are mathematical, which is not rigorous. How does the model perform on tasks from other subjects, such as MMLU or GPQA?

+ What is the difference between letting the model choose between CoT and TIR and selecting separate CoT and TIR models for reasoning? If a Math Model is used for CoT reasoning and a Code Model is used for TIR reasoning, would the performance improve, and would the method still be effective?

+ In Table 2, why is there no difference in performance between TATA and TATA-A200?

+ In Section 6.3, regarding the discussion on reinforcement learning, does it show that the TATA method does not have a scaling property and cannot be further improved through reinforcement learning? Does this indicate that the TATA algorithm has significant limitations in practical applications?

---

### Official Review · Reviewer_dWU5 · 2025-10-28

**Soundness:** 2
**Presentation:** 3
**Contribution:** 2
**Rating:** 4
**Confidence:** 4

**Summary:**

This paper introduces an adaptive fine-tuning framework that enables large language models to dynamically switch between Chain-of-Thought and Tool-Integrated Reasoning for mathematical problem solving, leveraging aptitude-aware data selection to align reasoning strategies with the model's inherent capabilities and achieving comparable accuracy and inference efficiency compared to existing methods.

**Strengths:**

- The paper proposes the TATA framework for mathematical reasoning tasks, enabling large language models to adaptively select between CoT and TIR strategies based on their inherent aptitude before reasoning.
- The paper is clearly written and easy to follow.

**Weaknesses:**

- The data construction process relies entirely on GPT-4o to rewrite CoT into TIR without any human verification, which may hinder faithful reproduction and fair comparison across studies.
- The framework determines the selection criteria using quantiles of performance differences between CoT and TIR, but this rule is heuristic and lacks deeper theoretical grounding or interpretability.
- The construction of the anchor set depends on clustering within a single embedding space, which may introduce representation bias and affect model aptitude assessment.
- Although the dataset construction process is explained, it would be more convincing to include concrete examples of data instances to help readers intuitively understand how the dataset differs from standard CoT or TIR training data.
- Compared with existing approaches, the improvements in both in-domain and out-of-domain benchmarks appear modest. Conducting statistical significance testing would help determine whether these gains are meaningful rather than due to random variation.

**Questions:**

- It would be helpful if the authors could further explain the rationale behind adopting the quantile-based heuristic in the decision rule.
- Could the authors provide more details on how the anchor-set construction mitigates potential bias from the embedding space?

---

### Official Review · Reviewer_abwp · 2025-10-30

**Soundness:** 3
**Presentation:** 3
**Contribution:** 3
**Rating:** 6
**Confidence:** 3

**Summary:**

This paper explores the use of LLMs in math reasoning and introduces an adaptive framework, TATA, that dynamically switches between CoT and TIR based on their respective strengths. The framework leverages a scoring mechanism, calculated from the accuracy of each mode, to guide the model’s decision-making process. Sufficient experiments demonstrate the superior performance of TATA, highlighting its effectiveness in reasoning strategies for mathematical tasks.

**Strengths:**

- The paper is well organized and presented, and the message is clearly conveyed.
- The paper is solving a relevant research problem and the proposed method is well motivated.
- The comparison of the proposed TATA with existing SOTA approaches is thorough, providing a detailed and deep analysis of its advantages.

**Weaknesses:**

- The proposed framework builds upon several well-established components that are already widely recognized in the field. Therefore, the contribution of this paper appears to be somewhat limited in novelty. The authors should clarify what aspects of their framework are unique and discuss the potential directions for future research that can make the approach more impactful.
- The framework switches between modes based on a score derived from accuracy, which does not capture the complexity of math problem-solving. Other factors such as task complexity, robustness, and reasoning depth should also be considered when selecting the mode. This may limit the generalizability of the model to complicated cases. Did the authors consider these factors?

**Questions:**

- The paper presents a thorough comparison between TATA and hybrid/ensemble models, however, the comparison is not consistent. Specifically, in Table 2, the accuracy is not compared with CoT and TIR using Qwen2.5-0.5B, while Table 10 includes such a comparison with LLaMA-3-8B. Can the authors clarify the design choice for these comparisons?
- Following the second point in Weakness, what are the other factors that might be important for switching between modes? Is there a plan to extend the framework to consider these aspects besides the accuracy?
- In the experiments, the TIR format is constructed based on CoT solutions. Can the authors clarify why the TIR responses were not used directly? What are the potential advantages or limitations of constructing TIR responses in this way, especially considering the differences between the CoT and TIR reasoning paradigms?

---

### Official Review · Reviewer_KMKn · 2025-11-03

**Soundness:** 2
**Presentation:** 2
**Contribution:** 2
**Rating:** 4
**Confidence:** 4

**Summary:**

The paper observes that existing SFT methods for math reasoning mostly stick to a single reasoning style—either Chain-of-Thought (CoT), which is efficient but sometimes numerically weak, or Tool-Integrated Reasoning (TIR), which is numerically reliable but slower. Prior work tried to combine them using external selectors or hand-crafted rules, but didn’t really let the LLM itself learn when to use which style. The authors propose TATA (Teaching LLMs According to Their Aptitude): given a dataset where each problem has both a CoT and a TIR solution, they evaluate the base model on an anchor set using each style, see which style the model is better at for that kind of query, and then build an SFT dataset tailored to that model’s “aptitude.” After SFT on this personalized data, the model is better at adaptively choosing CoT vs TIR at test time, and they show gains on several math benchmarks for both general LLMs and math LLMs.

**Strengths:**

1.The paper gives a good motivation: CoT and TIR succeed on different math subtypes (logic-heavy vs computation-heavy), so forcing one paradigm for all data is suboptimal.

2.Instead of a fixed routing heuristic or external selector, TATA makes the training data itself conditional on the base LLM’s aptitude.

3.They test on multiple math benchmarks and with different base models (general vs math-specialized), showing the method is not tied to a single model family.

**Weaknesses:**

1.A major contribution of this paper is the routing strategy for switching from text reasoning and tool use. But it has been naturally used in existing tool-augmented LLM that invokes tools when necessary in the reasoning Chain. So, it is hard to say the contribution of this paper is significant. Also, with only sft, the learned routing mechanism is not clearly well adapted into the LLM, but more likely the imitation of the data distribution. It should not be the optimal strategy for routing.

2.the requirements of dual-annotated data is also another bottleneck for the broaden application of this work. Such kinds of data is not easy to collect, and may also involve potential noise and misalignment due to different prompts and tool use settings.

**Questions:**

1.can we use RL instead of sft to learn the routing strategy?

2.will different prompts or tool use setting cause significant divergence in training results?

**Details Of Ethics Concerns:**

Please refer to the weakness and question part.

---

### Note · Authors · 2025-12-03

**Comment:**

We would like to thank all the reviewers and the AC for their time and effort in reviewing our work. We have decided to withdraw our submission.

**Withdrawal Confirmation:**

I have read and agree with the venue's withdrawal policy on behalf of myself and my co-authors.